Using large language models to create narrative events

Bartalesi Valentina valentina.bartalesi@isti.cnr.it 1
Lenzi Emanuele 1 2
De Martino Claudio 1
1 Institute of Information Science and Technologies “Alessandro Faedo”—ISTI, National Research Council of Italy (CNR) , Pisa , Italy
2 Department of Information Engineering (DII), University of Pisa , Pisa , Italy
Kong Xiangjie
Electronic publication date: 2024 Oct 22
Publication date: 2024
Volume: 10
Electronic Location ID: e2242
Received 2024 Apr 24; Accepted 2024 Jul 15
Copyright: ©2024 Bartalesi et al.
Copyright year: 2024
Copyright holder: Bartalesi et al.
License: This is an open access article distributed under the terms of the Creative Commons Attribution License, which permits unrestricted use, distribution, reproduction and adaptation in any medium and for any purpose provided that it is properly attributed. For attribution, the original author(s), title, publication source (PeerJ Computer Science) and either DOI or URL of the article must be cited.
License URL: https://creativecommons.org/licenses/by/4.0/

Keywords: Large language models, Narratives, Events, Semantic web, Digital humanities

Funding: ITSERR (Italian Strengthening of the ESFRI RI RESILIENCE) The MUR National Recovery and Resilience Plan The European Union-NextGenerationEU CRAEFT The European Union’s Horizon Europe research and innovation programme 101094349 This work was externally supported by ITSERR (Italian Strengthening of the ESFRI RI RESILIENCE), funded under the MUR National Recovery and Resilience Plan funded by the European Union-NextGenerationEU, and by CRAEFT, funded under grant agreement No 101094349 funded by the European Union’s Horizon Europe research and innovation programme. There was no additional external funding received for this study. The funders had no involvement in the study’s design, data collection and analysis, publication decision, or manuscript preparation.

==============================
Narratives play a crucial role in human communication, serving as a means to convey experiences, perspectives, and meanings across various domains. They are particularly significant in scientific communities, where narratives are often utilized to explain complex phenomena and share knowledge. This article explores the possibility of integrating large language models (LLMs) into a workflow that, exploiting the Semantic Web technologies, transforms raw textual data gathered by scientific communities into narratives. In particular, we focus on using LLMs to automatically create narrative events, maintaining the reliability of the generated texts. The study provides a conceptual definition of narrative events and evaluates the performance of different smaller LLMs compared to the requirements we identified. A key aspect of the experiment is the emphasis on maintaining the integrity of the original narratives in the LLM outputs, as experts often review texts produced by scientific communities to ensure their accuracy and reliability. We first perform an evaluation on a corpus of five narratives and then on a larger dataset comprising 124 narratives. LLaMA 2 is identified as the most suitable model for generating narrative events that closely align with the input texts, demonstrating its ability to generate high-quality narrative events. Prompt engineering techniques are then employed to enhance the performance of the selected model, leading to further improvements in the quality of the generated texts.

Introduction

A narrative is a method of articulating life experiences meaningfully (Schiff, 2012) and a conceptual basis of collective human understanding (Wertsch & Roediger, 2008). Humans use narratives, or stories, to explain and communicate abstract experiences, describe a particular point of view on some domain of interest, and share meanings among different communities (McInerny et al., 2014). A widely-held thesis in psychology to justify the centrality of narrative in human life is that humans make sense of reality by structuring events into narratives (Bruner, 1991; Taylor, 1992).

Maps have always geographically supported stories and storytelling, especially to represent the geospatial components of narratives, e.g., places cited in a story, territory’s influences on an author, geographic space descriptions that can be themselves the focus of a story (Caquard & Cartwright, 2014). On the other hand, as Korzybski pointed out in his General Semantics (Korzybski, 1933), there is a perceptual gap between the territory and its map in narratives. The perceptive cartographic challenge for a map is when it tries to represent also the life, emotions, reality, fiction, legends, and expectations associated with the described territory. Caquard & Cartwright (2014) outlines three storytelling forms for which maps provide spatial structure to enhance the storytelling experience on the places narrated in a story or on a story about a territory. These three forms are: oral, written, and audio-visual. Story maps are included in the third form and are computer science realisations of narratives based on interactive online maps enriched with text, pictures, videos, data, and other multimedia information. Overall, they are commercial platforms, inviting a diverse and non-expert user group to participate in the map-based story process to tell their own place-based stories (Kerski, 2015). In our opinion, story maps can be used both to enhance the perception of the geographic space cited in narratives and also to overcome Korzybski’s perceptual gap, enriching maps with media that communicate emotional messages associated with the described territory, e.g., digital audio/video material to describe the territorial complexity. We explored a specific domain (mountain territories and value chains) and proposed a general software solution for a semi-automatic transformation from text to narrative. In particular, we built a workflow to transform the data collected in an MS Excel file into structured story maps, whose textual content is finally organised into an interconnected knowledge graph, which can be exploited for story correlation analysis and new knowledge inferences (Bartalesi et al., 2023). The workflow utilizes natural language processing algorithms (Coro et al., 2017; Coro et al., 2021) to extract key terms (such as individuals, locations, organizations, and keywords) that hold significant importance in comprehending the text from the MS Excel file. Subsequently, it links these terms to Wikidata entries and, for locations, extracts the corresponding geographic coordinates. The workflow then generates a series of narrative events enriched with titles, descriptions, entities, coordinates, and multimedia data. Finally, it populates a semantic knowledge base modelled on an ontology designed for narrative formal representation (Meghini, Bartalesi & Metilli, 2021). This workflow is designed to be independent of specific application contexts and relies solely on open-source components. Currently, the entire workflow operates in an automated manner, with the exception of the event creation step. Indeed, the workflow can be initiated using the text collected in MS Excel files. In our opinion, the main current limit of our workflow is that it takes as input MS Excel or comma separate value (CSV) files only. Although many scientific communities use these formats to collect and organise data, much information is still gathered as raw texts.

In this article, we propose an approach for the event creation process to automate the workflow completely, taking raw textual data as input. To this aim, we decided to integrate the large language models (LLMs) into our workflow to explore their capability to split plain text narratives into events. In addition to the ability to divide the text into events, our main requirement is that the text associated with each event has to be as similar as possible to the text that the LLMs take as input. This requirement is motivated by the fact that usually, in the context of scientific communities, texts have been produced and reviewed by experts who ensure the reliability of the gathered information.

The article is structured as follows: Section Methodology describes the methodological steps we followed, including the task definition and the experimental setup, the requirements for model selection, the dataset used for the evaluation and the prompt engineering activity. Section Results presents the evaluation results by testing the prompt we defined with six different LLMs. In Section Discussion, we report some considerations about the obtained results. Finally, Section Conclusions offers final remarks and outlines future work.

Methodology

We adopted a bottom-up approach to integrate into our workflow a functionality that allows for the automatic creation of narrative events from raw text using large language models. Initially, we provided a conceptual definition of what we mean by event, comparing it with definitions found in Narratology, Artificial Intelligence, and natural language processing (NLP) literature. Subsequently, we experimented to select the best model from a set of LLMs, identifying a list of requirements the selected model has to meet. Finally, we created a small dataset to test the models and engaged in prompt engineering activities to achieve optimal performance. Based on the results obtained from this evaluation, a model was selected. Thus, the selected model was assessed on a larger dataset of narratives to verify its performance, and a prompt enhancement was performed. Due to the fact that narratives can belong to different literary genres (Girolimon, 2023; Conte & Most, 2015; Pavel, 2003), as they can be created on any topic, using any style or trope, and involving completely different character types, we considered fine-tuning the LLMs not useful in our domain of interest. Indeed, we aim to create an approach that is as general as possible and can be applied to any narrative genre.

All the methodological steps and their results are reported and described in the remainder of this section and in Section Results.

Task definition

The main prerequisite to define our task is to state what we intend as event. Indeed, several definitions are reported in the literature. For example, in the field of Narratology (Fludernik, 2009), which is the discipline devoted to the study of the narrative structure and the logic, principles, and practices of its representation, an event is defined as an occurrence taking place at a certain time in a specific location (Propp, 1968; Shklovsky, 1917). On the other hand, in the Artificial Intelligence (AI) literature, and in particular, in the Event Calculus theory (Kowalski & Sergot, 1986; Miller & Shanahan, 2002; Mueller, 2014), a generalized event is defined as a space–time chunk, in a context where events and objects are aspects of a physical universe with a spatial and temporal dimension. From a natural language processing point of view, in the context of the Automatic Content Extraction (ACE) 2005 evaluation campaign, an event is defined as “a specific occurrence involving participants” (Walker et al., 2005). The event triggers are verbs or nouns that most clearly express the core meaning of an event based on predefined event types.

In our research, we define an event as the fundamental component of a narrative. A narrative is conceptualized as a semantic network of events interconnected through semantic relationships. The identification of events within a textual narrative is carried out by the narrator, who determines the number of events to be created and assigns the corresponding text to each event. The textual content of each event is required to convey meaning and has to adhere to syntactical completeness, avoiding incomplete sentences. Consequently, our definition aligns more closely with those found in the Narratology and Event Calculus, as opposed to the definition used in the ACE 2005 campaign, which emphasises linguistic aspects more. The task we want to perform consists of using LLMs to split plain text narratives into events, considering our own definition of event.

Requirements for model selection

To perform the task defined in the section above, we need to identify the language model that is able to achieve the best results. In particular, we collected a list of requirements that the model has to satisfy to be integrated into our workflow, that is:

1. we need an open-source model since we aim to integrate it as a service into our open-source-oriented workflow;

2. the model does not have to require powerful hardware to guarantee the repeatability of the entire workflow also with limited hardware resources available (i.e., a desktop PC with an AMD Ryzen 7 5800X3D 8-core 16-thread processor, Nvidia GeForce RTX 4070 SUPER 12 GB video card, and 32 GB of system RAM);

3. the model has to produce outputs without generating new text or modifying the input text, i.e., it should not generate any new words or sentences or delete text;

4. once the model is integrated into our workflow as a service, it should be used by different users to perform the same task; thus, we have to consider the output coherency, i.e., the outputs of the model obtained in the different runs have to be as equal as possible;

5. the model has to identify syntactic consistent events. For consistent events, we intend the events that do not start or finish with incomplete sentences.

To identify an LLM that can meet all these requirements, we evaluated two tools designed for easily and locally testing various LLMs: Ollama (2023) and LM Studio (2023). Ollama is an open-source software offering a simple API and a library of pre-built models, which can significantly reduce developers’ time and effort. When conversing with a model, developers typically need to compose prompts in the specific syntax format required by that particular model. Ollama simplifies this process by automatically handling various prompt formats, allowing developers to test different models using a universal syntax. LM Studio is a desktop application used to install and test open-source LLMs. Although this application is not open-source, it offers a user-friendly interface that allows users to explore the capabilities of open-source LLMs. However, unlike Ollama, LM Studio does not provide the ability to control and manage model responses using a programming language (Gerganov, 2023). Due to this limitation of LM Studio, we decided to choose Ollama as tool to test LLMs.

Considering Requirement 1 about the need to use open-source models, but especially Requirement 2 regarding the limited hardware resources we want to use to guarantee the repeatability of this experiment, we opted to test only the open-source and smaller LLMs provided by Ollama. Specifically, we evaluated the following models: LLaMA 2 7B, Mistral 7B, Mistral 7B:text, orca-mini, phi-2, and Openchat. Notice that we selected Mistral 7B:text to assess the performance of a specialized LLM version designed for simple text completion. When accessible, we evaluated the chat versions of the aforementioned models for efficiency reasons. By chat, we refer to the conversational mode of the LLM, which will henceforth assume this definition unless stated otherwise. All these models also meet our hardware requirements. Specifically, the hardware available for this task cannot effectively handle models with a complexity of 13B and above within a reasonable execution time. The technical specifications of the hardware we used for testing the LLMs are provided below:

• AMD Ryzen 7 5800X3D 8-core 16-threads processor

• Nvidia GeForce RTX 4070 SUPER 12 GB video card

• 32 GB of system RAM

Dataset

The context length refers to the number of tokens a language model can process at a given time. It is predetermined in transformer-based models. This parameter is crucial as it limits the number of words a language model can handle at once. Additionally, in chat applications, the context length determines how much of the previous conversation is retained (Riccio, 2023). We tested the selected LLMs using the Ollama framework on a dataset comprising five plain text narratives (Bartalesi, De Martino & Lenzi, 2024c), all written in English. We opted to use a small dataset to ensure a correct evaluation of the LLM outputs. The narratives in this dataset cover various topics, styles, tropes, and character types (e.g., fictional and non-fictional), have similar lengths and were written by scholars or machines. They include a narrative about a medieval (1414) journey of an Italian humanist authored by a scholar in Latin literature from the University of Pisa (Mele, 2022) (this narrative also incorporates poetic elements), a story about the giant squid blending legend and science written by a CNR scientist expert in biological computation modelling (Coro et al., 2015), an account of the Battle of Britain in 1940 and a science fiction novel both created by ChatGPT and a narrative about the discovery and study of the prehistoric coelacanth fish, as reported in an article by Weinberg (2013). The narratives comprise the following number of tokens: the medieval journey consists of 1,451 tokens; the giant squid of 1,524 tokens; the Battle of Britain of 864 tokens; the fiction novel of 1,163 tokens; and the discovery of the coelacanth fish of 1,438 tokens, for an average token count of 1,288.

Prompt engineering

Our objective is to identify the model capable of achieving the best result in segmenting plain text narratives into events. To accomplish this goal, we conducted a prompt engineering activity to determine the most effective prompt. Initially, we defined two different user prompts (Ramlochan, 2024). In the first prompt, we reported our notion of event. In the second one, we asked to take as input a narrative and divide it into events without adding or deleting any token. However, after some tests, we opted to remove the second prompt. This choice is determined by the tendency of the models to forget the information provided in the first prompt while performing the second one. This behaviour is due to the limited capacity of the 7B models, which struggle to maintain long-term coherence during a conversation. When used with more than one prompt, they have difficulty in remembering the specific information provided previously. This can lead to a loss of context and the production of less coherent or relevant responses. In other words, the memory capacity of language models is limited, and this can affect their ability to sustain a coherent answer across multiple turns.

Therefore, we passed to a single user prompt. In this prompt, we provided the definition of event and requested the narrative to be segmented into events without adding or deleting any tokens. Unfortunately, the results were unsatisfactory due to the limited complexity of the 7B models, which hinders their understanding of the event definition. As a result, we opted to remove the event definition from the user prompt, allowing different LLMs to interpret this concept based on their training. At the same time, we decided to switch from the user prompt to the system prompt to optimise the model performance.

Results

Through a Python script (Bartalesi, De Martino & Lenzi, 2024i), we passed our dataset (i.e., a list of strings) as input to all the selected LLMs applying the prompt described above (see Section Methodology). We evaluated Requirement 3, 4 and 5 using the selected measures reported in the following. For some of these measures, we report the average values. The averages were calculated by adding the values of each LLM iteration and dividing by the total number of outputs, i.e., three iterations for five narratives for a total of 15 outputs. To evaluate the difference between the source text and the one produced by the models (Requirement 3), we calculated the average of the Jaccard similarity coefficient values (Real & Vargas, 1996) (“Jaccard (avg)” label in the following tables). Furthermore, we calculated the average number of events that are present in the outputs of the models (“No events (avg)” label in the following tables). If this number is equal to 1, it means that the input text was not split into events. Finally, we calculated the number of failed prompts that represent iterations which produce an output that was not divided into events (“Failed prompts” label in the following tables). The number of failed prompts was automatically checked through a Python script (Bartalesi, De Martino & Lenzi, 2024j) in all the outputs. The script counts the number of outputs that do not contain any newline character, which is used to separate events by LLMs. If the number of failed prompts is high, this means that the model mainly produced outputs that are a copy of the input text without splitting it into events.

To evaluate Requirement 4 regarding the coherency of the LLM outputs, we stored all the different textual outputs produced in three iterations, excluding duplicates, for each narrative. Then, we compute the average for each model (“Output coherency (avg)” label in the following tables). The range of possible values is between 1 and 3, with extremes included.

Finally, we calculated the percentage of the events that are inconsistent in terms of syntactic structure, i.e., the events that include incomplete sentences (Requirement 5). The syntactic consistency was checked manually on the result of a Python script (Bartalesi, De Martino & Lenzi, 2024h). This script retrieves the sentences of the LLM outputs that don’t finish with one of the following punctuation marks: full stops, question marks, or exclamation marks (“Broken Sentences” label in the following tables).

The settings of all the tested LLMs are the same, i.e., a temperature of 0.01 to reduce output randomness and a context length of 4,096 tokens to ensure that the entire input is correctly processed.

The results of this evaluation are reported in Table 1. The outputs of the LLMs are available on a public-access Figshare repository (Bartalesi, De Martino & Lenzi, 2024d). We decided to discard Mistral 7B:text since it repeated the same answer without stopping, going in a loop. We also discarded phi-2 since it never stopped, and it has extrinsic hallucinations (Ji et al., 2023), producing answers completely different from the ones expected.

Table 1 Performances of different models on the dataset of five narratives.

Model	Jaccard (avg)	No events (avg)	Failed prompts	Output coherency (avg)	Broken sentences	
llama2:7b-chat-q4_0	0.31	10.25	0	2.40	3	
mistral:7b	0.35	6.45	0	2.0	9	
openchat:7b-v3.5-q4_0	1.0	4.20	12	1.75	0	
orca-mini:7b	0.70	5.17	10	1.64	0	

The Jaccard values of Openchat and orca-mini could be misleading if they are not evaluated together with the average number of events and the failed prompt number. Indeed, the values of these two last parameters (columns 3 and 4 of Table 1) highlight the tendency of these models to replicate the input text as output, creating a few events and having a high number of failed prompts. On the other hand, LLaMA and Mistral show lower Jaccard values but create narratives richer in events, and no prompt fails. Regarding the output coherency (column 5), the table shows that orca-mini achieves the best results, followed by Openchat. Mistral and LLaMA obtain the worst results. Finally, the results of the syntactic consistency evaluation of the LLM outputs (last column of the table) show that orca-mini and Openchat achieve the best result, while Mistral and LLaMA report the worst results. Indeed, after manually checking the results, we noticed that these two last models have difficulty in handling the correct punctuation when sentences are reported between quotation marks. Specifically, there is a reversal of the quotation marks with the trailing period, i.e., instead of having the correct form (”.), the model reports the wrong form (.”). Below, we provide one example for LLaMA and another for Mistral. Starting from the input text: Bishop Erik Pontoppidan describes the Kraken as the world’s largest animal (a circumference of 2.5 km) in his work, “The natural history of Norway (1752; 1752-1753)”., LLaMA produces the following output: Bishop Erik Pontoppidan describes the Kraken as the world’s largest animal (a circumference of 2.5 km) in his work “The natural history of Norway.”. The same punctuation issue appears in the output of Mistral, which, taking as input text: Roman naturalist Pliny the Elder (first century AD), in his work “Natural history”, described a giant squid, which had a body “as large as a barrel” and tentacles reaching 9.1 m in length., returns as output: Roman naturalist Pliny the Elder described a giant squid with a body “as large as a barrel” and tentacles reaching 9.1m in length in his work “Natural history.”.

We tried to work on the context length to enhance the Jaccard values of some models and reduce the number of failed prompts of others models. We adopted an iterative approach by progressively reducing the token number of the input texts. We segmented each narrative collected in our dataset into multiple inputs, and then we passed them to each model. We performed several experiments with texts of different lengths. We noticed that shorter text provided better results in terms of avoiding text removal or new text generation in the LLM outputs. Ultimately, following a bottom-up approach, we determined that a limit of 500 tokens for the input texts represents a good compromise between Jaccard values and event-splitting performance (Bartalesi, De Martino & Lenzi, 2024e). The evaluation results for each model are reported in Table 2. We made the LLM outputs available on a public-access Figshare repository (Bartalesi, De Martino & Lenzi, 2024f).

Table 2 Performances of different models on the dataset of shorter narratives.

Model	Jaccard (avg)	No events (avg)	Failed prompts	Output coherency (avg)	Broken sentences	
llama2:7b-chat-q4_0	0.60	9.62	0	1.60	0	
mistral:7b	0.66	10.00	0	1.47	3	
openchat:7b-v3.5-q4_0	1.0	1.0	15	1.35	0	
orca-mini:7b	0.81	4.00	12	1.36	0	

As shown in Table 2, even if the highest Jaccard coefficient value is reported for Openchat, it is associated with an average number of events equal to 1, which means that the input text was not split into events. Furthermore, the number of failed prompts is 15/15. Orca-mini follows with a lower Jaccard. However, the number of created events is still low. These values demonstrate that despite using shorter input texts, the performance of these two models remains poor for our task. On the contrary, the LLaMA and Mistral produced better performances in terms of the number of created events and failed prompts. However, the Jaccard values (0.60 and 0.66, respectively) are lower than those of Openchat and orca-mini.

In comparison with Mistral, LLaMA shows a lower output coherence value but achieves the best results in terms of syntactic consistency. Overall, compared to Table 1, we can see that the division into shorter narratives helps the models to enhance their performances.

We performed one more experiment in order to improve the performance of the models. In particular, we defined a new system prompt asking for splitting the input narratives with a length of 500 tokens into paragraphs instead of events. In this way, we wanted to test if the concept of paragraph is similar or not to the concept of event for the LLMs. As reported in Table 3, using this new prompt, the Jaccard of LLaMA and orca-mini improve consistently, while Mistral gets slightly better. The value of Openchat does not change in comparison to that reported in Table 2. In parallel, the average number of events decreases for LLaMA and Mistral and remains equal to 1 for Openchat; also, the value of orca-mini gets worse, from 4.0 to 1.0. The increase in Jaccard values and the simultaneous decrease in the average number of created events demonstrate that, with this new prompt, LLaMA and Mistral generate events that are more similar to the input text and richer in content. However, the Jaccard values and the number of events produced by Openchat remain unchanged, while those provided by orca-mini indicate that this model is unable to split the narratives into events. The results of the failed prompts align with those reported in Table 2 for all models, with a worsening observed for orca-mini, increasing from 12 to 15. Additionally, coherence in the produced outputs improves across all models, with Mistral also exhibiting enhanced syntactic consistency, while this value was already zero for the other models. We published the LLM outputs on an open-access Figshare repository (Bartalesi, De Martino & Lenzi, 2024g). On the basis of these results, we can infer that the notion of paragraph is very close to the notion of event for the LLMs we used. This claim is supported by the definition of paragraph as reported in the Cambridge Dictionary (Cambridge University, 2024). This dictionary defines a paragraph as “a short part of a text, consisting of at least one sentence and beginning on a new line. It usually deals with a single event, description, idea, etc.”. At the same time, the notion of paragraph is also close to our conceptual definition of event. Thus, from now on, we refer to event as a synonym of paragraph and to this prompt as main prompt.

Table 3 The performance of the LLMs using the main prompt in which event is a synonym of paragraph.

Model	Jaccard (avg)	No events (avg)	Failed prompts	Output coherency (avg)	Broken sentences	
llama2:7b-chat-q4_0	0.86	4.86	0	1.40	0	
mistral:7b	0.69	5.14	0	1.27	0	
openchat:7b-v3.5-q4_0	1.0	1.0	15	1.20	0	
orca-mini:7b	1.0	1.0	15	1.16	0	

Due to the results reported in Tables 2 and 3, we have determined LLaMA to be the most suitable model for our task. Indeed, LLaMA shows the best results for most of the evaluation measures: it achieves the highest Jaccard coefficient, generates a satisfactory number of events, avoids prompt failures, and produces syntactically consistent sentences. To assess that LLaMA is really the best model for this task, we decided to perform another assessment, testing all the models on a larger dataset.

Evaluation on a large dataset

We created a larger dataset of narratives to verify the results of our first assessment, for which we used only five narratives. This new dataset is composed of a total of 124 narratives, including the five ones from the previous dataset (Bartalesi, De Martino & Lenzi, 2024a). All the narratives have a length of about 500 tokens and are written in English. The length was defined based on the results reported in Table 2. They are composed manually, starting from Wikipedia pages. The narrative text is extracted from different sections of Wikipedia pages, but it was manually reviewed to eliminate bibliographic references, phonetic alphabet signs and links to other resources. Eleven of the 124 narratives include words or sentences in different languages (i.e., French, Spanish, Italian, Latin, and German). The dataset consists of narratives about various topics, from geographical descriptions to biographies, from historical events to book plots. We evaluated the performances of the selected LLMs on this new dataset.

For this evaluation, the settings of all the tested LLMs are the same as those used in the assessment conducted on the small dataset, i.e., 0.01 temperature and a context length of 4,096 tokens.

The values reported in Table 4 confirm that LLaMA is the best model for our task, producing a reasonable Jaccard value average and an acceptable number of events. On the contrary, Mistral fails to maintain syntactically correct sentences in several cases, while Openchat and orca-mini fail to split the original text into events most of the time. The outputs generated by the LLMs can be found on a publicly accessible Figshare repository (Bartalesi, De Martino & Lenzi, 2024b).

Once LLaMA was selected as the best model for our task, we tested different quantizations of LLaMA on our larger dataset to verify which version produces the best results.

The results reported in Table 5 show that the version of LLaMA with 8-bit quantization produces better results than the versions with 4 and 5-bit quantization. For this reason, we decided to adopt LLaMA 2 7B with 8-bit quantization for the further experiments reported in this article.

Prompt enhancement

Once we identified LLaMA 2 7B 8-bit quantization as the best model for our task, we tried to enhance our prompt to improve the values of the Jaccard coefficient, unchanging the values of the other parameters. Following a bottom-up approach, we carried out two experiments in which we defined two different Jaccard coefficient thresholds below which an enhanced prompt has to be applied. In the first experiment, we set the threshold to 0.90. In the second experiment, the threshold was set at 0.98.

The enhanced prompt was designed by using special characters as markers (OpenAI, 2024). Specifically, we defined a system prompt containing a title, delimited by three hashtag symbols (i.e., #), which emphasizes that the prompt contains the instructions that the model has to execute. The instructions are the same as reported in our main prompt.

Table 4 Comparison among different models on the large dataset.

Model	Jaccard (avg)	No events (avg)	Failed prompts	Output coherency (avg)	Broken sentences	
llama2:7b-chat-q4_0	0.89	7.23	0	1.38	0	
mistral:7b	0.83	6.46	0	1.43	21	
openchat:7b-v3.5-q4_0	0.99	1.20	348	1.01	0	
orca-mini:7b	0.98	1.26	344	1.06	0	

Table 5 Comparison among different quantizations of LLaMA on the large dataset.

Model	Jaccard (avg)	No events (avg)	Failed prompts	Output coherency (avg)	Broken sentences	
llama2:7b-chat-q4_0	0.89	7.24	0	1.35	0	
llama2:7b-chat-q5_0	0.91	8.77	0	1.41	4	
llama2:7b-chat-q8_0	0.92	7.70	0	1.33	0	

We decided to test the enhanced prompt on the LLaMA best output only. We define the best output as the one with the highest Jaccard value, producing no failed prompt and at least two events (a value that ensures that no copy and paste has been made from the input) and contains no broken sentences.

Table 6 shows the results of the evaluation of the enhanced prompt on the large dataset. The first row reports the performance of the main prompt, the second row shows the results using the enhanced prompts with the Jaccard threshold set to 0.90, and the third row shows the results using the enhanced prompts with the Jaccard threshold set to 0.98.

Table 6 Performance of LLaMA 2 with the main prompt (first row), enhanced prompt with 0.90 Jaccard threshold (second row) and with 0.98 Jaccard threshold (third row).

Model	Jaccard (avg)	No events (avg)	Failed prompts	Output coherency (avg)	Broken sentences	
llama2:7b-chat-q8_0	0.93	7.61	0	1.17	0	
llama2:7b-chat-q8_0 - 0.90 threshold	0.937	7.57	0	1.14	0	
llama2:7b-chat-q8_0 - 0.98 threshold	0.939	7.53	0	1.14	0	

The results show that the main prompt performed well on the large dataset. The average value of the Jaccard coefficient is 0.93, with an average of created events for each narrative of 7.61. The average coherence of the outputs is 1.17, and no failed prompt or broken sentences are reported. The results of the evaluation with the application of the enhanced prompt show a slight increase in the Jaccard coefficient value (+0.01) and a slight decrease in the average created events (−0.04) and output coherency (−0.03). The remaining results are the same as the main prompt.

Figure 1 shows the Jaccard values for each of 124 narratives, using 0.90 (vermilion orange line) and 0.98 (black line) as thresholds. The graph shows that only a few narratives (nine) are increased in their Jaccard value using a 0.98 threshold.

Figure 1 Comparison of Jaccard values of the 124 narratives with 0.90 and 0.98 thresholds.

Discussion

Based on our experience and the definitions reported in the literature, the concept of event in narratives is vague, and its proper definition depends on the application domain and the narrator. For example, some narrators interpret an event as a spatiotemporal occurrence with associated dates and locations; others intend events as descriptions of a subject where the time is the one in which the event is narrated, and the space is absent. Carrying out the experiment described in this article, we notice that the interpretation of the event given by the LLMs we tested is similar to the definition of paragraph reported by the Cambridge Dictionary, for which “a paragraph usually deals with a single event”. This is the rationale on the basis of which we defined a prompt asking to split raw textual narratives into paragraphs, which in our experiment are synonyms of events.

The presented experiment aims to identify a smaller LLM that is mainly able to create a list of events starting from raw text narratives with the request to maintain the original texts without generating new text or reformulating the source sentences. We decided to evaluate only smaller LLMs since our computational resources, such as memory, storage, and processing power, are very limited. We are aware that our request is naive because we do not want to use LLMs as models for generating text, but we aim to reuse reliable text produced by experts or scholars. LLMs excel in producing text that sounds natural and coherent. However, they occasionally can make mistakes or invent information. Indeed, LLMs can generate information that is not grounded in factual evidence and is potentially influenced by biases inherent in its transformer architecture. Summarising, the model fabricates facts, posing challenges in domains where factual precision is crucial. Disregarding consistent factual accuracy poses a danger in a world where combating misinformation and disinformation relies on accurate and dependable information. Scientific communities collect a lot of reliable and accurate data that are rich in narratives if organized and presented following a formal narrative structure. In the context of this experiment, we want to assess if LLMs can be used to organise these data into narrative events, guaranteeing their trustworthiness at the same time. We compared the performances of six different LLMs on a small dataset, using different prompts, and then we selected the best one for our task, i.e., LLaMA 2. To ensure the feasibility of this result, we have also evaluated all the models on a more extensive narrative dataset, which has confirmed LLaMA 2 as the best model for our task. Then, we tried to enhance the system prompt to improve the Jaccard values. When we set a Jaccard threshold to 0.90, the enhanced prompt allows a slight increase in the Jaccard values (with an average of about 0.1 for 124 narratives), while the execution time is about 100 min. Setting a Jaccard threshold of 0.98, the increment of the Jaccard values is smaller (0.002 for 124 narratives), and the execution time increases from 100 to 158 min. We can deduce that setting a Jaccard threshold of 0.98 is not convenient when comparing execution time and obtained results.

The final consideration pertains to the possibility of refining the LLaMA outputs to enhance their similarity with the input texts. This step becomes necessary due to the conversational nature of the chat version of LLaMA, which often responds to input by echoing parts of the question. Additionally, there are instances where LLaMA may opt to number the events or append a brief explanatory message at the end of the division. Cleaning the text of any formatting discrepancies helps mitigate pitfalls arising from the diversity of texts. This cleaning phase holds particular significance and sensitivity in meeting the requirement of preserving the originality of the input narrative. The primary challenge in this process lies in distinguishing words and punctuation marks originating from the input text from those added by LLaMA. Indeed, there’s a risk of inadvertently eliminating words, or of creating broken sentences because of the addition of wrong punctuation. This risk is also critical for calculating the Jaccard similarity coefficient.

Conclusions

We have presented an experiment in which we used LLMs to split raw text data into narrative events. The main requirement of our experiment is that the text associated with each event has to be as similar as possible to the text that the LLMs take as input. This requirement is motivated by the fact that the data produced by scientific communities have often been reviewed by experts who ensured their reliability. We compared the performances of six different LLMs on a small dataset, using different prompts, and then we selected the best one for our task, i.e., LLaMA 2. To ensure the feasibility of this result, we have evaluated all the models also on a larger narrative dataset, which has confirmed LLaMA 2 as the best model for our task. Then, we tried to enhance the selected prompt, and we reported the corresponding results. Although our task aims to use LLMs in a somewhat naive manner by limiting their text generation capabilities in favour of greater reliability of the produced texts, we have found that LLaMA 2 is still able to generate excellent results for this task.

In the future, we intend to expand our evaluation to encompass a broader range of narratives, spanning additional topics and genres. This broader evaluation will provide deeper insights into the robustness of LLaMA 2 across different narrative domains. Moreover, we intend to conduct an in-depth analysis of fine-tuning LLaMA to assess and enhance its performance within a dataset of narratives covering a specific topic (e.g., narratives about cultural heritage objects). Finally, we intend to integrate the experiment described in this article into the workflow we are developing to create story maps starting from raw texts as a service. The textual content of these story maps is organised into an interconnected knowledge graph of narrative events. Thus, in our context, automating the generation of narrative events is crucial in terms of reducing the time of data processing for scientific community experts. Our experiment demonstrates that LLMs, and in particular LLaMA 2, can be used to reach this goal.

Additional Information and Declarations

Competing Interests

Author Contributions

Data Availability

The authors declare there are no competing interests.

Valentina Bartalesi conceived and designed the experiments, performed the experiments, analyzed the data, prepared figures and/or tables, authored or reviewed drafts of the article, and approved the final draft.

Emanuele Lenzi performed the experiments, analyzed the data, performed the computation work, prepared figures and/or tables, authored or reviewed drafts of the article, and approved the final draft.

Claudio De Martino performed the experiments, analyzed the data, performed the computation work, prepared figures and/or tables, authored or reviewed drafts of the article, and approved the final draft.

The following information was supplied regarding data availability:

The Python script for LLM testing is available at Figshare: aimhdhgroup (2024). Python script for LLMs testing. figshare. Software. https://doi.org/10.6084/m9.figshare.25585683.v3.

The Python script for prompt fail checking is available at Figshare: aimhdhgroup (2024). Python script for prompt fail checking. figshare. Software. https://doi.org/10.6084/m9.figshare.25585722.v3.

The Python script for consistency checking is available at Figshare: aimhdhgroup (2024). Python script for consistency checking. figshare. Software. https://doi.org/10.6084/m9.figshare.25585824.v3.

The dataset of 124 narratives is available at Figshare: aimhdhgroup (2024). Dataset of 124 Narratives. figshare. Dataset. https://doi.org/10.6084/m9.figshare.25562400.v4.

The dataset of 5 narratives is available at Figshare: aimhdhgroup (2024). Dataset of 5 Narratives. figshare. Dataset. https://doi.org/10.6084/m9.figshare.25562406.v3.

The dataset of 5 short narratives is available at Figshare: aimhdhgroup (2024). Dataset of 5 Short Narratives. figshare. Dataset. https://doi.org/10.6084/m9.figshare.25562433.v2.

The dataset of five narratives split (∼1200 tokens) into events is available at Figshare: aimhdhgroup (2024). Dataset of five narratives split (1200 tokens) into events. figshare. Dataset. https://doi.org/10.6084/m9.figshare.26046436.v3.

The dataset of five short narratives (∼500 tokens) split into events is available at Figshare: aimhdhgroup (2024). Dataset of five short narratives (500 tokens) split into events. figshare. Dataset. https://doi.org/10.6084/m9.figshare.26046445.v3.

The dataset of five short narratives (∼500 tokens) split into paragraphs is available at Figshare: aimhdhgroup (2024). Dataset of five short narratives (500 tokens) split into paragraphs. figshare. Dataset. https://doi.org/10.6084/m9.figshare.26046448.v3.

The dataset of 124 narratives (∼500 tokens) split into paragraphs is available at Figshare: aimhdhgroup (2024). Dataset of 124 narratives (500 tokens) split into paragraphs. figshare. Dataset. https://doi.org/10.6084/m9.figshare.26046457.v3.

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
