# Peer review of "Using large language models to create narrative events"

_PeerJ Computer Science, doi:10.7717/peerj-cs.2242_

## Round 0.1 · original submission · Major Revisions

Your submission is not ready for acceptance currently, but it can be evaluated again if it can be revised properly. Please improve it according to the comments.

·

Basic reporting

This paper proposes creating narrative events using large language models(LLMs), an approach that explores the possibility of integrating LLMs into semantic workflows using the raw textual data as input while maintaining overall performance and integrity.

In the “Methodology” section, from lines 82 to 83: “Finally, we created a small dataset to test the models and engaged in prompt engineering activities to achieve optimal performance. Based on the results obtained from this evaluation, a model was selected.”. LLMs have the best(or more accurate) evaluation result when the model is finetuned in the specific domain datasets. Based on the “Dataset” section, the narrative dataset is not particularly large. Were there any considerations of mixing this dataset with general samples?
In the “Requirements for model selection” section, from lines 115 to 116: “the model does not have to require powerful hardware to guarantee the repeatability of the entire workflow also with limited hardware resources available”. It would be helpful to have a more detailed definition of “limited hardware resources”. What is the final environment to run the LLM in the automated workflow? As the previous section indicates it seems the model takes in raw text input from the website, does this mean it will need to run in the browser background? Or offline inference works too in this case.
Thanks for providing a clear introduction to the dataset and token numbers.
The paper maintains a clear structure and provides a comprehensive list of literature studies on narratology.

Experimental design

In the “Dataset” section, it would be helpful to also cite the sources and why these particular datasets were chosen. In addition, it would be helpful to include more context on “each narrative has about 1200 tokens”.
In the “Evaluation on a large dataset” section, from lines 265 to 266: “All the narratives have a length of about 500 tokens and are written in English.”. Based on the testing dataset token length per narrative, does this mean the evaluation dataset contains a much shorter context window?
In the “Discussion” section, there’s a grammar error in line 323: “Occasionally, they may err or make information up.”.

Validity of the findings

The findings of the paper are considered valid as it is supported by experiments conducted on open datasets. For a more accurate presentation of the research work, please address the comments above.

Additional comments

Thanks for the opportunity to review the manuscript. Please address the comment above.

Cite this review as

·

Basic reporting

Clarity and Language:
The manuscript is clearly written and well-structured. The language is professional and accessible, with minimal grammatical errors. The introduction effectively sets the context for the study and clearly states the research objectives.
Literature Review:
The literature review is comprehensive and covers relevant studies on large language models (LLMs) and event segmentation in text narratives. It provides a solid foundation for understanding the significance of the study.
Figures and Tables:
The manuscript includes detailed tables summarizing the performance of each model, which are clear and well-organized. However, figures illustrating the experimental setup or examples of segmented narratives would enhance the manuscript further.
References:
References are relevant and appropriately cited throughout the manuscript.

Experimental design

Study Design:
The study design is robust, with a clear explanation of the methods used to evaluate the LLMs. The choice of models (LLaMA 2 7B, Mistral 7B, etc.) is justified, and the rationale for selecting these models is well-explained.
Dataset:
The dataset comprises diverse narratives from various genres, which is appropriate for evaluating the models’ ability to segment different types of text. The size of the dataset (narratives around 1200 tokens) is sufficient for robust testing.
Experimental Procedure:
The experimental procedure is thoroughly described, including the use of different prompt structures and context lengths. The methods for measuring performance (Jaccard similarity, number of events created, failed prompts, etc.) are clearly defined.
Reproducibility:
The detailed description of the experimental setup, including hardware specifications, ensures that the study can be replicated by other researchers.

Validity of the findings

Data Analysis:
The data analysis is rigorous, with multiple iterations conducted to ensure the robustness of the findings.
The use of multiple metrics (Jaccard similarity, coherency, syntactic consistency) provides a comprehensive assessment of model performance.
Results:
The results are clearly presented, with well-organized tables summarizing the performance of each model. The findings are consistent with the stated objectives and hypotheses.
Interpretation:
The interpretation of the results is logical and supported by the data. The manuscript discusses the strengths and limitations of each model, providing a balanced view of their performance.
Conclusions:
The conclusions are well-supported by the findings and align with the study’s objectives. The discussion includes practical implications and suggestions for future research.

Additional comments

Strengths:
The study addresses an important area of research in natural language processing. The methodology is rigorous, and the findings are robust and well-presented. The manuscript is well-written and easy to follow.
Areas for Improvement:
1.Include examples of segmented narratives from each model to illustrate the differences in performance.
2.Address the punctuation issues observed in models like LLaMA and Mistral. Include specific examples of the punctuation issues and discuss possible ways to address them. For example, "LLaMA 2 7B often misplaces quotation marks, leading to syntactically incorrect sentences. An example of this is: 'The scientist said 'This discovery is groundbreaking.' instead of 'The scientist said, 'This discovery is groundbreaking.''
3.The description of the dataset is somewhat brief. While it mentions that the narratives are around 1200 tokens and come from diverse topics, it does not specify the number of narratives used or the exact nature of these narratives
4.While the manuscript mentions output coherency and syntactic consistency, it does not provide specific metrics or methods used to evaluate these aspects.
5.The manuscript mentions hardware limitations but does not elaborate on how these limitations might have impacted the results or the choice of models.

Cite this review as

---

## Round 0.2 · Minor Revisions

I'm happy to inform you that your article is potentially accepted. However, you need to revise the work carefully according to the comments of the reviewers.

·

Basic reporting

The authors added further explanation of the definition of narratives and the reason for creating a method that's considered more general.

Experimental design

The authors have answered all questions in the last review about the dataset and discussions of results and added details in the paper accordingly.

Validity of the findings

N/A

Cite this review as

·

Basic reporting

The revised manuscript employs clear and professional English throughout. The responses to reviewers' comments are concise and add clarity to the paper. The introduction and background sections effectively establish the context of the study. The literature review is comprehensive, referencing relevant and recent studies. The manuscript's structure adheres to PeerJ standards, with well-organized sections that enhance readability. The manuscript includes references to the raw data. The additions and revisions have improved the paper's clarity and logical flow. The figures included in the manuscript are relevant, high-quality, However the Figure 1 Axis labels and legends are hard to read and could be formatted better for visibility.

Experimental design

The manuscript presents original primary research that falls within the scope of the journal. The study addresses an important aspect of narrative segmentation in language model.
The research question is clearly defined, relevant, and meaningful. The investigation is rigorous and adheres to high technical and ethical standards. The methods are well-described, ensuring the reliability of the findings.The methods section is detailed and thorough, providing sufficient information for replication. The manuscript includes specific examples and clarifications in response to reviewers' comments, enhancing the reproducibility of the study.

Validity of the findings

The manuscript encourages meaningful replication, providing detailed methods and raw data, with clear rationale and benefits for replication stated. The underlying data are robust, statistically sound, and well-controlled, addressing reviewers' concerns effectively. Conclusions are well-stated, directly linked to the original research question, and supported by the results and data provided. The manuscript revisions enhance clarity and coherence in presenting the conclusions.

Additional comments

The authors have addressed all the reviewer's comments comprehensively. The revisions have significantly improved the manuscript's clarity, depth, and transparency. The additional data and examples provided enhance the paper's robustness and reproducibility. The manuscript is now well-structured and meets the editorial criteria effectively.

Overall, the manuscript is of high quality and makes a valuable contribution to the field of narrative segmentation in language models. I recommend it for publication with minor revisions to ensure all the additional data and clarifications are fully integrated and presented in a clear, accessible manner.

Cite this review as

---

## Round 0.3 · accepted · Accept

Thanks to the authors for your efforts to improve the work. This version has satisfied the reviewers successfully.